# Hemolysis induced by Left Ventricular Assist Device is associated with proximal tubulopathy

Tristan de Nattes[1]*, Pierre-Yves Litzler[2,3], Arnaud Gay[2], Catherine Nafeh-Bizet[2], Arnaud François[4], Dominique Guerrot[1,3]

1 Nephrology–Kidney Transplant Unit, Rouen University Hospital, Rouen, France, 2 Thoracic and Cardiovascular Surgery Department, Rouen University Hospital, Rouen, France, 3 INSERM U1096, Rouen University Hospital, Rouen, France, 4 Department of Pathology, Rouen University Hospital, Rouen, France

* tristan.de-nattes@chu-rouen.fr

**Data Availability Statement:** The datasets used and/or analyzed during the current study are available from the figshare database (DOI: 10.6084/m9.figshare.12958325).

## Abstract

### Background

Chronic subclinical hemolysis is frequent in patients implanted with Left Ventricular Assist Device (LVAD) and is associated with adverse outcomes. Consequences of LVADs-induced subclinical hemolysis on kidney structure and function is currently unknown.

### Methods

Thirty-three patients implanted with a *Heartmate II* LVAD (Abbott, Inc, Chicago IL) were retrospectively studied. Hemolysis, Acute Kidney Injury (AKI) and the evolution of estimated Glomerular Filtration Rate were analyzed. Proximal Tubulopathy (PT) groups were defined according to proteinuria, normoglycemic glycosuria, and electrolytic disorders. The Receiver Operating Characteristic (ROC) curve was used to analyze threshold of LDH values associated with PT.

### Results

Median LDH between PT groups were statistically different, 688 IU/L [642–703] and 356 IU/L [320–494] in the "PT" and "no PT" groups, respectively p = 0.006. To determine PT group, LDH threshold > 600 IU/L was associated with a sensitivity of 85.7% (95% CI, 42.1–99.6) and a specificity of 84.6% (95% CI, 65.1–95.6). The ROC's Area Under Curve was 0.83 (95% CI, 0.68–0.98). In the "PT" group, patients had 4.2 [2.5–5.0] AKI episodes per year of exposure, versus 1.6 [0.4–3.7] in the "no PT" group, p = 0.03. A higher occurrence of AKI was associated with subsequent development of Chronic Kidney Disease (CKD) (p = 0.02) and death (p = 0.05).

### Conclusions

LVADs-induced subclinical hemolysis is associated with proximal tubular functional alterations, which in turn contribute to the occurrence of AKI and subsequent CKD. Owing to renal toxicity of hemolysis, measures to reduce subclinical hemolysis intensity as canula

**Funding:** The authors received no specific funding for this work.

**Competing interests:** The authors have declared that no competing interests exist.

**Abbreviations:** AKI, Acute Kidney Injury; AUC, Area Under Curve; CKD, Chronic Kidney Disease; GFR, Glomerular Filtration Rate; LDH, Lactate Dehydrogenase; LVAD, Left Ventricular Assist Device; PT, Proximal Tubulopathy; ROC, Receiver Operating Characteristic; SCD, Sickle Cell Disease.

position or pump parameters should be systematically considered, as well as specific nephroprotective therapies.

## Introduction

Heart transplantation remains the gold standard treatment for end stage heart failure, but due to the lack of grafts, the use of Left Ventricular Assist Devices (LVAD) has continuously increased during the last decades. LVADs improve survival compared to any conventional medical treatment [1] with an overall survival of 83% at 12 months and 46% at 5 years and more than 25000 patients have already been implanted [2].

Changes in renal function after LVADs implantation have been reported, but critical analysis of renal function data is frequently limited, leading to contradictory results [3]. Brisco *et al* reported that 22% of patients improved their glomerular filtration rate (GFR) by 49%, but this improvement was described as generally transient [4]. Hasin *et al* found that 68% of patients with a pre-implantation GFR $< 60$ mL/min/1.73m$^2$ showed a significant improvement of GFR 3 and 6 months after surgery [5]. These data suggest that patients with LVADs implanted in a chronic setting first have an improved kidney function due to better hemodynamic parameters but that secondarily, LVADs could lead to an accelerated decline in GFR due to parenchymal lesions [6]. Moreover, LVADs are associated with Acute Kidney Injury (AKI) with an incidence between 7.4% and 9% [7, 8].

Chronic subclinical hemolysis, which is frequently due to pump thrombosis, can be one of the complications of mechanical circulatory assist devices. Monitoring of the LDH value is considered as a pertinent biomarker and has to be regularly monitored [9]. In other circumstances, such as Sickle Cell Disease (SCD) and paroxysmal nocturnal hemoglobinuria, hemolysis has been established as a cause of proximal tubulopathy [10]. It has also been shown by Muslem *et al* that pre-operative proteinuria before LVADs implantation is associated with mortality and hemodialysis at one year [11]. Furthermore, the need of hemodialysis after LVADs implantation is associated with an increased risk of mortality, underlying the impact of renal function in LVADs patients survival [12]. In the present study, we hypothesized that subclinical hemolysis induced by LVADs is a cause of tubular injury and associated with subsequent decrease in kidney function.

## Materials and methods

### Study population

All patients with a *Heartmate II* Left Ventricular Assist Device (Abbott, Inc, Chicagon IL) implanted in Rouen University Hospital between June 2006 and November 2017 were screened by retrospective review. The research procedures were in accordance with the French law and approved by the ethics committee of Rouen, France (CERNI Nr E2018-58). In line with this committee, and considering that the data were analyzed anonymously, consent were not obtained.

Clinical and LVADs data were obtained from the patient's medical record. Biological data concerning hemolysis, proximal tubulopathy (PT) and renal function were collected from the hospital laboratory. Data were collected every week during the first month, every three months during the first year, and then every six months until the end of follow-up. Baseline MDRD eGFR was calculated with the last stable plasma creatinine available before hospitalization for *Heartmate II* implantation.

Urine analyses were considered only when the corresponding blood analyses were performed on the same day. By retrospective review, alternative etiologies for hemolysis were excluded: There were no hereditary disorders (spherocytosis, sickle cell anemia, G6PD deficiency), Coombs' tests were negative in all cases, and hemolytic anemia could not be explained by drugs' toxicity, infectious agent or chronic autoimmune disease. Similarly, etiologies for proximal tubulopathy were systematically excluded.

Follow-up was stopped when LVADs were definitely explanted: heart transplantation or death. In our study, no patient presented a reversible cardiomyopathy.

### Hemolysis and proximal tubulopathy

We defined subclinical hemolysis as recommended by INTERMACS guidelines [13]: serum lactate dehydrogenase (LDH) level greater than two and one-half time the upper limit of the normal range (i.e $> 700$ IU/L in our laboratory), without clinical symptoms of hemolysis. Average LDH was calculated for each patient, based on blood samples collected every week during the first month, every three months during the first year, and then every six months until the end of follow-up.

To define PT, we considered proteinuria/creatininuria ratio between 0.15 and 1.5 g/g and presence of glycosuria without hyperglycemia as major criteria, and presence of aseptic leukocyturia, renal potassium waste (hypokalemia $< 3.5$ mmol/L with inadequate kaliuresis) and hypophosphatemia (serum phosphate $< 2.7$ mg/dL) as minor criteria (Table 1).

Patients with two major criteria on two urine and blood samples, with at least one minor criterion were included in the Proximal Tubulopathy group (PT group). The other patients were considered as no Proximal Tubulopathy group (no PT group). Due to the fact that diabetic patients may have transient normoglycemic glycosuria related to diabetes, analyzes were also performed after exclusion of this major criterion in diabetic patients.

### Acute kidney injury and chronic kidney disease

We defined AKI as recommended by the KDIGO 2012 guidelines [14]: increase in plasma creatinine $> 26.5$ μmol/L within 48 hours, or increase in plasma creatinine $> 1.5$ fold baseline, which is known or presumed to have occurred within the prior 7 days. Then, we calculated a ratio with the number of AKI per year of Heartmate *II* exposure. Chronic Kidney Disease (CKD) was defined as recommended by the KDIGO 2012 guidelines.

### Statistical analysis

Quantitative data were presented as median [25th-75th percentile]. The differences between PT groups were compared with Mann-Whitney U and Chi-square tests. A sensitivity/specificity Receiver Operating Characteristic (ROC) curve test was used to determine the best cut-off

**Table 1. Definition of major and minor criteria of Proximal Tubulopathy (PT).** Owing to the fact that diabetic patients may have normoglycemic glycosuria, analyzes were also performed after exclusion of this major criterion in diabetic patients.

| | |
|---|---|
| Major criteria | proteinuria/creatininuria ratio between 0.15 and 1.5 g/g |
| | glycosuria without hyperglycemia |
| Minor criteria | aseptic leukocyturia |
| | hypokalemia < 3.5 mmol/L with inadequate kaliuresis |
| | serum phosphate < 2.7 mg/dL |
| Proximal tubulopathy | two major criteria, on at least two urine and blood samples, + at least one minor criterion. |

value of the LDH value associated with PT. Statistical analyses were conducted using Graph-Pad Prism version 6.04, GraphPad Software, La Jolla California USA, and confidence level was set at 0.05.

## Results

Forty-four patients were implanted, four patients were excluded because their survival was less than 60 days and six patients were excluded from the Proximal Tubulopathy analysis because of insufficient biological data. Clinical characteristics of the remaining 33 patients are summarized in Table 2. At baseline, no patient had other cause of hemolysis or sign of tubulopathy.

In our unit, all patients were treated by anticoagulation therapy with a therapeutic INR range between 2.0 and 2.5. As previously published, we do not use antiplatelet therapy in order to reduce major bleeding events [15, 16].

### Renal function after *Heartmate II* implantation

Before *Heartmate II* implantation, 14 patients (42.4%) had a CKD with a baseline MDRD eGFR < 90 mL/min/1.73m$^2$ and 9 patients (27.7%) had an eGFR < 60 mL/min/1.73m$^2$. Two patients (6.0%) had proteinuria at baseline: one patient in each group (proteinuria/creatininuria ratio was 0.6 g/g for both of them).

At baseline, median plasma creatinine was 109 μmol/L [76–145], MDRD eGFR = 63 mL/min/1.73m$^2$ [45–89]. At the end of follow-up, plasma creatinine was 102 μmol/L [86–149], with a eGFR of 60 ml/min/1.73m$^2$ [43–81]. Twenty-eight patients (84.8%) had an eGFR < 90mL/min/1.73m$^2$, 17 patients (51.5%) an eGFR < 60 mL/min/m$^2$, and 2 patients (6.1%) an eGFR < 30mL/min/m$^2$. No patient underwent chronic hemodialysis.

**Table 2. Clinical characteristics of patients.** Values are medians [25th-75th percentile].

| Characteristics | N = 33 (%) |
|---|---|
| Age (year) | 63 [58–70] |
| Female sex, n (%) | 6 (18.1) |
| Body-mass index | 24.91 [22.98–30.01] |
| Diabetes, n (%) | 7 (21.2) |
| Hypertension, n (%) | 23 (69.7) |
| Causes of heart failure, n (%) | |
| Ischemic | 21 (63.6) |
| Dilative | 11 (33.3) |
| Myocarditis | 1 (3.0) |
| Estimated GFR < 90 mL/min/1.73m$^2$, n (%) | 14 (42.4) |
| Estimated GFR < 60 mL/min/1.73m$^2$, n (%) | 9 (27.2) |
| Plasma creatinine at the implantation (μmol/L) | 109 [76–145] |
| eGFR at the implantation (mL/min/1.73m$^2$) | 63 [45–89] |
| HMII support duration (days) | 693 [355–1120] |
| Transplanted, n (%) | 18 (54.5) |
| Dead, n (%) | 10 (30.3) |
| Bridge to transplantation | 3 (30.0) |
| Destination therapy | 7 (70.0) |
| Alive with HMII, n (%) | 5 (12.8) |
| Bridge to transplantation | 3 (60.0) |
| Destination therapy | 2 (40.0) |

Twenty-six patients (78.8%) had AKI between hospitalization and LVAD implantation, without requiring hemodialysis. Nineteen patients (57.6%) had AKI the first month after *Heartmate II* implantation. Six patients (18.2%) had AKI between the first month and hospital discharge. Among these patients, 7 underwent acute hemodialysis. There was no difference at baseline between patients who required acute hemodialysis and others. More, the need of acute hemodialysis after LVADs implantation did not influence the presence or absence of PT (S1 Table). The median total number of AKI was 4.0 [1.0–8.0], which corresponded to 2.5 AKI/year of exposure [0.5–4.4].

### Hemolysis and proximal tubulopathy

Twenty-two patients (56.4%) developed significant proteinuria (mean proteinuria/creatininuria ratio = 0.97 g/g in proteinuric patients). There was no hematuria. Seventeen patients (43.6%) had normoglycemic glycosuria. Seven patients (21.1%) were considered as "Proximal Tubulopathy" and twenty-six (78.8%) as "no Proximal Tubulopathy". All patients were classified in their PT group before the first year. Kidney function before implantation was not different between tubulopathy groups.

LDH evolution during the first 18 months is presented in Fig 1. There was a significant difference between the two groups: in "PT" group, median LDH was 688 [643–703] and 356 [320–494] in the "no PT" group, p = 0.006 (Fig 2). The statistical significance persisted after exclusion of normoglycemic glycosuria as major criterion of PT in diabetic patients (p = 0.03).

Six patients had a LVAD thrombosis: 4 patients (15.4%) in the "no PT" group and 2 patients (28.6%) in the "PT" group. All patients were classified in their PT group before occurrence of LVAD thrombosis, and all patients except one died after the pump thrombosis. Three other patients required a pump exchange, all classified in the "no PT" group

### Acquired von Willebrand disease

In "PT" group, 3 patients (42.3%) had symptomatic acquired von Willebrand syndrome, versus 14 patients (53.8%) in "no PT" group. All patients had received blood transfusions: 21 [12–36] transfusions per patient in the "PT" group and 29 [16–54] transfusions per patient in the "no PT" group during the entire follow-up period, p = 0.5.

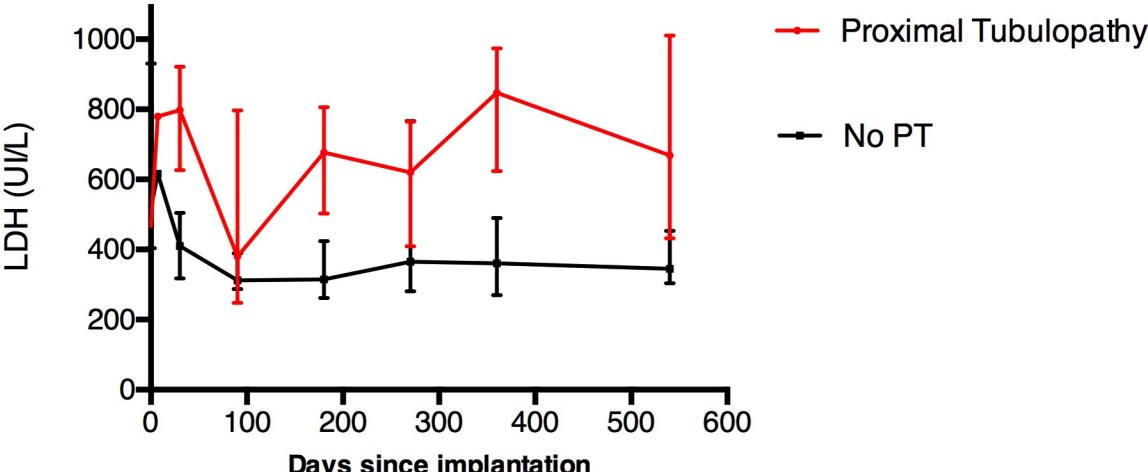

**Fig 1. Evolution of LDH value.** Median value of LDH (25th– 75th percentile) according to time since *LVAD* implantation.

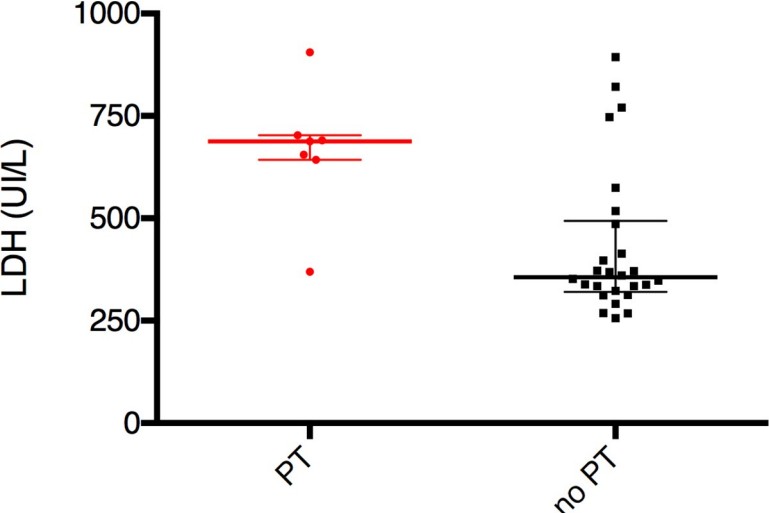

**Fig 2. Median value of LDH.** Median value of LDH (25th - 75th percentile) according to Proximal Tubulopathy (PT), p = 0.006.

There was no difference of LDH value between patients with versus without symptomatic acquired von Willebrand syndrome secondary to the *Heartmate II*, or according to other clinical characteristics (Table 3).

## Sensitivity and specificity of LDH cut-off value to determine PT group

The area under curve (AUC) of mean LDH value of each patient for the diagnosis of PT was 0.83 (95% CI, 0.68–0.98). A cut-off of 600 IU/L for mean LDH was associated with a 85.7% sensitivity (95% CI, 42,1–99,6) and 84.6% specificity (95% CI, 65.1–95.6). A cut-off of 700 IU/L was associated with a 28.6% sensitivity (95% CI, 3.7–8.0) and a 84.6% specificity (95% IC, 65.1–95.6). A cut-off of 500 IU/L was associated with a 85.7% sensitivity (95% CI, 42,1–99,6) and a 76.9% specificity (95% IC, 56.3–91.0) (Fig 3).

## Acute kidney injury

Incidence of AKI episodes per year of exposure was statistically different between "PT" and "no PT" groups, 4.2 [2.5–5.0] and 1.6 [0.4–3.7] AKI episodes per year of exposure, respectively, p = 0.03 (Fig 4). There was no association between number of AKI and baseline eGFR (p = 0.5).

Patients who died had 3.4 [2.6–4.7] AKI/y versus 1.6 [0.5–4.6] AKI/y in patients who had been subsequently transplanted (p = 0.05).

## Chronic kidney disease

Based on the last stable plasma creatinine available, groups defined by the end-of-follow-up MDRD eGFR regardless of PT statue differed significantly in terms of number of AKI per year of exposure. The group with an eGFR between 90 and 60 mL/min/1.73m$^2$ had a median of 0.5 [1.3–2.4] AKI/y versus 3.3 [1.5–4.7] AKI/y in the group with an eGFR between 60 and 30 mL/min/1.73m$^2$ and 5.4 [2.2–5.8] AKI/y in the group with an eGFR < 30 mL/min/1.73m$^2$ (p = 0.001) (Fig 5). A higher occurrence of AKI was therefore associated with subsequent development of CKD.

**Table 3. Characteristics of patients according to their Proximal Tubulopathy (PT) group.** Values are medians [25th-75th percentile].

| | PT, n = 7 (21.1%) | No PT, n = 26 (78.8) | p |
|---|---|---|---|
| Baseline characteristics | | | |
| Age (year) | 71 [61–72] | 62 [57–66] | 0.07 |
| Diabetes, n (%) | 2 (28.6) | 5 (19.2) | 0.6 |
| HbA1c (%) | 7.8 [7.2–8.4] | 7.2 [7.0–8.0] | 0.5 |
| Hypertension, n (%) | 5 (71.4) | 18 (69.2) | 0.9 |
| Plasma creatinine at the implantation (µmol/L) | 113 [86–137] | 106 [72–150] | 0.6 |
| eGFR (mL/min/1.73m$^2$) | 57 [48–83] | 65 [43–96] | 0.6 |
| Biological data | | | |
| • Hemoglobin (g/dL) | 10.9 [9.9–11.7] | 11.8 [9.5–13.0] | 0.3 |
| • Total bilirubin (µmol/L) | 19.0 [13.0–44.0] | 19.0 [12.8–31.3] | 0.9 |
| • Albumin (g/dL) | 32.0 [30.7–34.0] | 29.3 [25.5–32.0] | 0.2 |
| Blood pressure at the implantation (mmHg) | | | |
| • Systolic | 110 [84–115] | 100 [88–120] | 0.9 |
| • Diastolic | 70 [50–77] | 70 [55–90] | 0.5 |
| Cause of heart failure, n (%) | | | |
| • Ischemic | 4 (57.1) | 17 (65.4) | 0.7 |
| • Dilative | 3 (42.9) | 8 (30.8) | 0.5 |
| • Myocarditis | 0 | 1 (3.8) | |
| INTERMACS classification, n (%) | | | 0.5 |
| • 1 | 2 (28.6) | 5 (19.2) | |
| • 2 | 2 (28.6) | 13 (50.0) | |
| • 3 | 1 (14.3) | 5 (19.2) | |
| • 4 | 2 (28.6) | 2 (7.7) | |
| • 5–7 | 0 | 1 (3.8) | |
| Indication | | | 0.2 |
| • Destination therapy, n (%) | 3 (42.9) | 5 (19.2) | |
| • Bridge to transplantation, n (%) | 4 (57.1) | 21 (80.8) | |
| Temporary mechanical circulatory support, n (%) | | | 0.05 |
| • Impella | 0 | 9 (34.6) | |
| • ECMO | 6 (85.7) | 13 (50.0) | |
| Resternotomy, n (%) | 1 (14.3) | 4 (15.4) | 0.9 |
| ICD implantation, n (%) | 5 (71.4) | 16 (61.5) | 0.6 |
| Follow-up characteristics | | | |
| HMII support duration (days) | 511 [332–1091] | 715 [359–1186] | 0.9 |
| LDH | 688 [643–703] | 356 [320–494] | 0.006 |
| Acquired von Willebrand syndrome, n (%) | 3 (42.3) | 14 (53.8) | 0.6 |
| Transfusion (n) | 21 [12–36] | 29 [16–54] | 0.5 |
| Device or driveline infection | 6 (85.7) | 20 (76.9) | 0.6 |
| AKI (n of episode/year) | 4.3 [2.5–5.0] | 1.6 [0.4–3.7] | 0.05 |
| Plasma creatinine at the end of follow-up | 141 [62–152] | 100 [87–149] | 0.6 |
| eGFR (mL/min/1.73m$^2$) | 46 [42–118] | 64 [43–80] | 0.6 |
| End of follow-up, n (%) | | | |
| • Transplanted | 3 (42.9) | 15 (57.7) | 0.7 |
| • Dead | 4 (57.1) | 6 (23.1) | 0.2 |
| ○ Destination therapy | 3 (75.0) | 4 (66.7) | |
| • Alive with HMII | | 5 (19.2) | |

*(Continued)*

**Table 3.** (Continued)

| | PT, n = 7 (21.1%) | No PT, n = 26 (78.8) | p |
|---|---|---|---|
| ○ Destination therapy | | 1 (20.0) | |

ECMO: Extracorporeal Membrane Oxygenation. ICD: Implantable Cardioverter Defibrillators. HMII: Heartmate II. LDH: Lactate Dehydrogenase. AKI: Acute Kidney Injury. eGFR: estimated Glomerular Filtration Rate.

## Discussion

Despite the sample size, this study shows that in patients with *Heartmate II* device, chronic subclinical hemolysis is frequent and associated with renal outcomes, especially functional hallmarks of proximal tubular injury. Second, considering that LDH value is a commonly used marker of subclinical hemolysis, we propose a LDH threshold of 600 IU/L as an alarm signal for PT. Third, patients with LVADs-related PT present an increased occurrence of AKI. Finally, regardless of the presence of PT, patients who had the more AKI episodes had the worst renal function at the end of the follow up.

### Hemolysis and proximal tubulopathy

To date, there is no validated score for the diagnosis of Proximal Tubulopathy. Owing to its physiological function [17], the typical features of PT are low-molecular weight proteinuria [18], glycosuria and phosphaturia. Besides these signs, several indicators of Proximal Tubular

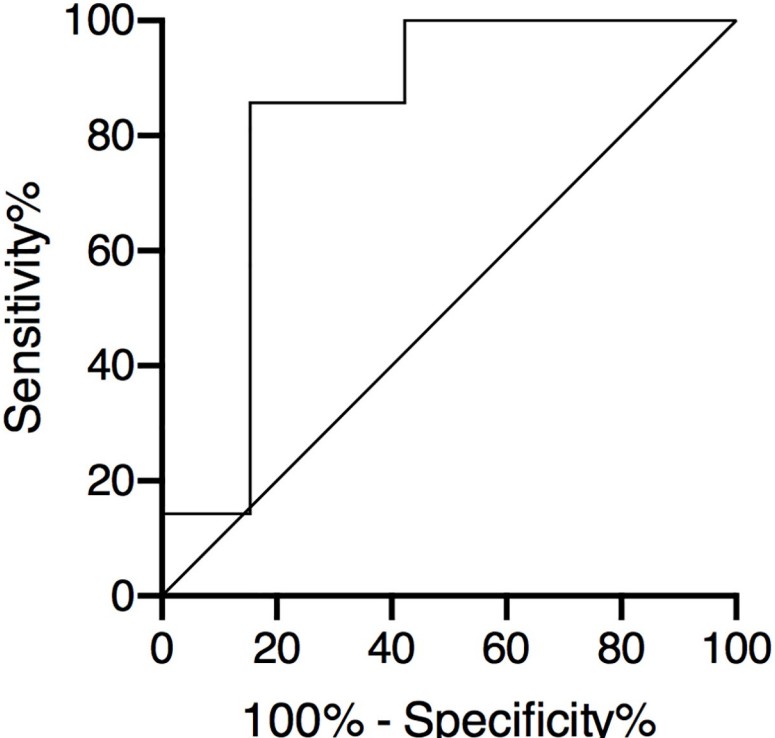

**Fig 3. ROC curve of LDH value for the diagnosis of proximal tubulopathy.** ROC AUC = 0.83 (95% CI, 0.68–0.98). Cut-off of 600 IU/L for median LDH among follow-up was associated with a 85.7% sensitivity (95% CI, 42,1–99,6) and a 84.6% specificity (95% CI, 65.1–95.6).

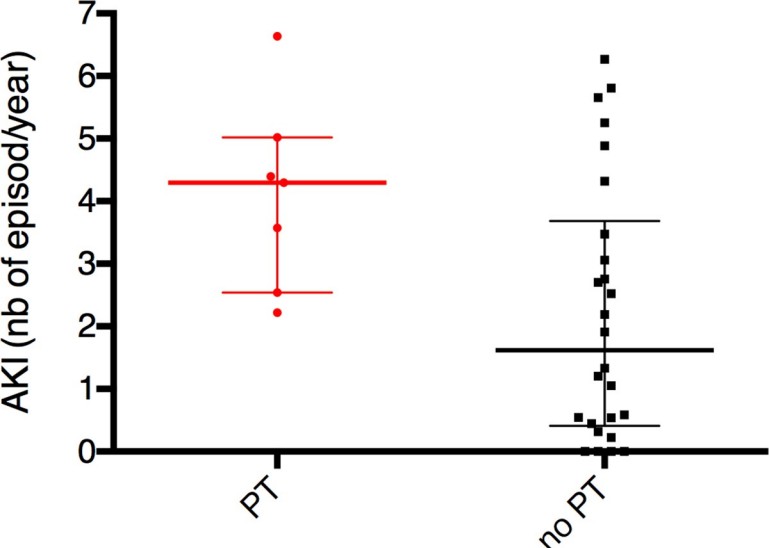

**Fig 4. AKI and proximal tubulopathy.** Number of AKI per year of exposure (median and 25th-75th percentile) to *LVAD* according to Proximal Tubulopathy (PT), p = 0.03.

Dysfunction are validated, mainly in management guidelines of patients infected by HIV who are particularly exposed to drug-induced PT, especially secondary to Tenofovir [19, 20]. Owing to that, red flags in the follow-up of these patients are: hypokaliemia, low serum bicarbonate, hypophosphatemia with abnormal fractional excretion of phosphate [21], abnormal fractional excretion of uric acid, glycosuria [22], aminoacidura and urinary B2 microglobulin [23]. These signs are validated in other diseases as Lowe syndrome [24, 25], drug-toxicities [26], and have been recently used as a cornerstone for the diagnosis of PT in the scope of COVID [27].

Due to its retrospective design, it has been necessary to choose, among those listed, biological elements for which the laboratory techniques did not change across years and which are

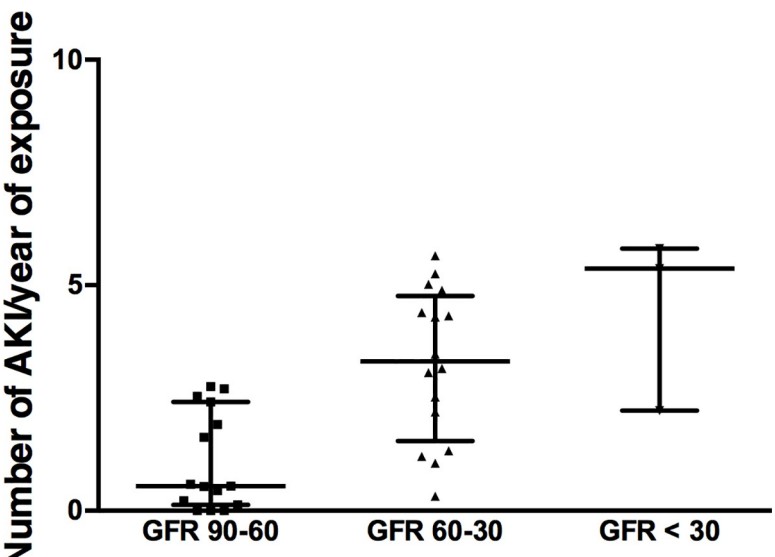

**Fig 5. Number of AKI per year of exposure (median and 25th-75th percentile) according to MDRD eGFR at the end of the follow-up, p = 0.001.**

widely performed in daily monitoring of LVAD patients. Consequently, we selected protein-uria and glycosuria as major criteria considering that these signs are more specific of tubular alterations. In order to differentiate proteinuria of glomerular versus tubular origin, we consid-ered usual hallmarks of tubular injury, i.e a low abundance around 1 g/day as compared to glo-merular proteinuria which is supposed to be higher than 2 g/day [18]. Moreover, we paid particular attention to hematuria, which is a classical sign of glomerular lesions, and therefore associated with glomerular proteinuria. In our study, no patient had hematuria, which is another argument for a proteinuria from tubular origin.

Owing to the fact that hypokalemia, serum phosphate and aseptic leukocyturia may reflect other issues in patients with LVAD (e.g diarrhea, malnutrition, treated urinary tract infection. . .), we made the choice to consider them as minor criteria. At least, specific signs as aminoaciduria or urinary B2 microglobulin were not routinely performed in our unit and were consequently excluded from the analysis. Finally, biological data were collected only in stable periods and patients were classified in the PT group only when two samples several weeks apart revealed the same findings.

Several diseases can induce renal siderosis. In paroxysmal nocturnal hemoglobinuria, there are reports of AKI due to hemosiderosis after acute hemolytic episodes [28], but also chronic tubular toxicity with nephrosiderosis attributed to chronic hemolysis. Moreover, patients with repeated acute hemolytic episodes had interstitial fibrosis which could be an indirect conse-quence of renal hemolysis toxicity, causing CKD in 20% of patients [29]. In SCD, it has been shown that tubular iron accumulation is correlated with severity of hemolysis and that hemo-lysis is associated with proteinuria and renal involvement [30]. There are several reports of AKI and nephrosiderosis after acute hemolysis in patients with mechanical heart valves. In 1990, 33 autopsies of patients with mechanical valves were performed. Nephrosiderosis was correlated with the delay since surgery and associated with number of AKI. Mean plasma cre-atinine was also correlated with severity of siderosis [31].

LVAD implantation can lead to chronic subclinical hemolysis, which is frequently the con-sequence of pump thrombosis and is statistically associated with higher mortality [32]. LDH is therefore commonly monitored and an increase over 700 IU/L (2.5 fold normal) is predictive of pump thrombosis and cerebrovascular accident [33]. In our study, we also found an associa-tion between the severity of hemolysis, estimated through LDH value, and PT which is the first step towards kidney damage: AKI and CKD. We found a threshold of LDH value which should alarm the clinician on an increased risk for kidney disease in patient with LVADs. Indeed, AUC of ROC has a value of 0.83 (95% CI, 0.68–0.98). Using a threshold of 600 IU/L of LDH, sensitivity and specificity were 85.7% (95% CI, 42.1–99.6) and 84.6% (95% CI, 65.1–95.6), respectively. Taken together, this data supports the use of LDH as a simple monitoring tool for the risk of subsequent chronic tubular alterations.

Interestingly, difference in hemolysis intensity was present since the first month after LVAD implantation, and patients were classified into their PT group before the twelfth month post implantation. These observations suggest that tubular injury occurs early and that severe short-term hemolysis can be an important factor in renal outcomes. Identification of individ-ual risk factors explaining renal sensitivity to hemolysis and improved management of hemo-lysis in early post-operative period could be of major interest.

## LVADs and acute kidney injury

Several cases of AKI after acute hemolysis have been reported. In our study, we examined occurrence of AKI in stable and chronic subclinical hemolysis. We found an association between PT and the number of AKI, suggesting that chronic proximal tubular functional

alterations can impair kidney adaptability. Although we did not identify a direct association between PT and CKD, which could be explained by the low number of patients, patients with worst renal function at the end of follow-up had more episodes of AKI than patients with normal or mildly impaired renal function. This highlights the need to limit all factors associated with renal aggression, e.g. hemolysis and other nephrotoxic agents. A recent study confirms that AKI is frequent, in approximately 70% of patients, and is associated with mortality one year after LVADs' implantation [34]. However, our work is the first considering the number of AKI throughout all exposure time and the association with CKD and death, confirming that occurrence of AKI represents an important long-term issue for patients with LVADs support.

## Therapeutic targets: How to limit subclinical hemolysis and renal toxicity?

In association with general nephroprotective recommendations, limiting subclinical hemolysis and its renal toxicity could be a therapeutic approach. Shear stress due to LVADs' pump leads to blood trauma and activates several pathways: von Willebrand Factor degradation, platelet and coagulation activations. Occurrence of these complications depends in part on the pump's physical properties, and recommendations have been proposed to avoid them [35, 36]. Development of more hemocompatible pumps such as magnetic centrifugal-flow pump improves subclinical hemolysis, which is associated with maintenance of endothelial function compared to continuous axial-flow devices [37, 38]. It also has been shown that shallower outflow canula angulation is associated with reduced thrombogenicity [39]. Moreover, the pump rotation speed influences hemodynamic flow and shear stress [40] and the choice of an optimal echo-guided rotation speed is of major interest [41].

In addition, avoiding renal toxicity of hemolysis has been studied. Several diseases can cause chronic hemolysis, including SCD, paroxysmal nocturnal hemoglobinuria and mechanical valves. Although renal involvement in these diseases is generally the consequence of several pathological pathways, the role of chronic hemolysis in renal toxicity is predominant, which suggests a potential benefit of specifically targeted therapeutic options.

In a SCD mouse model, a treatment with human haptoglobin leads to an increased expression of Heme Oxygenase-1 and decreased iron deposition in the kidney [42]

Due to the involvement of Heme Oxygenase-1 in several models of AKI and CKD, this may become an interesting therapeutic option in hemolysis-related kidney injury [43, 44]

In the scope of endothelial dysfunction due to free Hb and free heme, Kasztan *et al.* evaluated efficiency of ambrisentan, a selective endothelin-A receptor antagonist, in a murine model of SCD. Endothelin-A receptor is activated by endothelin-1, leading to inflammation and vasoconstriction. With daily administration of ambrisentan, long-term tubular and glomerular injuries were prevented [45]. The same team has initiated a clinical trial in which renal function of patients with SCD and treated with ambrisentan will be evaluated (NCT02712346). Validations of these strategies could be of major interest in the scope of LVAD.

## Limitations

Our study has several limitations due to its retrospective design. Diagnosis of PT lacks the reliability of specific investigations of tubular function. Although this score has not been previously validated, it was built with classical hallmarks of proximal tubular dysfunction and we believe it provides reliable information. Since the study population was stratified post-hoc based on the PT score, potential confounding cannot be excluded. In addition, a validation would theoretically have been required to ascertain the ROC analysis performed. Although this study has a monocentric design, the main characteristics of our population, including

intensity of hemolysis, are comparable to those reported in larger studies. Malnutrition is a frequent issue in chronic heart failure, and could be a potential bias in the estimation of GFR based on plasma creatinine. Moreover, renal function was impaired on an acute mode for most of patients at the time of implantation, and similarly, was frequently unstable at the end of follow-up. It should be noted that most studies on LVADs and renal function faced the same methodological issues regarding GFR.

## Conclusion

To our knowledge, this is the first study to demonstrate that LVADs-related hemolysis is associated with renal functional impairment. A LDH value > 600 IU/L is a relevant marker of PT in this population. Moreover, PT is associated with the risk of AKI. Since subclinical hemolysis and the mechanisms leading to its renal toxicity are potentially improvable, we believe that these results may have significant implications for patients implanted with LVADs, and deserves investigations in a larger confirmation cohort.

## Supporting information

**S1 Table. Characteristics of patients according to the need of acute hemodialysis after LVADs implantation.** Values are medians [25th-75th percentile]. ECMO: Extracorporeal Membrane Oxygenation. LDH: Lactate Dehydrogenase. AKI: Acute Kidney Injury. (DOCX)

## Author Contributions

**Conceptualization:** Tristan de Nattes, Pierre-Yves Litzler, Catherine Nafeh-Bizet, Arnaud François, Dominique Guerrot.

**Data curation:** Tristan de Nattes, Arnaud Gay, Arnaud François, Dominique Guerrot.

**Formal analysis:** Tristan de Nattes.

**Investigation:** Tristan de Nattes, Pierre-Yves Litzler, Arnaud François, Dominique Guerrot.

**Methodology:** Tristan de Nattes, Pierre-Yves Litzler, Dominique Guerrot.

**Supervision:** Dominique Guerrot.

**Validation:** Pierre-Yves Litzler, Dominique Guerrot.

**Writing – original draft:** Tristan de Nattes, Pierre-Yves Litzler, Dominique Guerrot.

**Writing – review & editing:** Tristan de Nattes, Pierre-Yves Litzler, Arnaud Gay, Catherine Nafeh-Bizet, Arnaud François, Dominique Guerrot.

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
