## [Decision Letter · Decision Letter 0]

7 Jul 2020

PONE-D-20-15799

Hemolysis induced by Left Ventricular Assist Device is associated with proximal tubulopathy.

PLOS ONE

Dear Dr. de Nattes,

Thank you for submitting your manuscript to PLOS ONE. After careful consideration, we feel that it has merit but does not fully meet PLOS ONE’s publication criteria as it currently stands. Therefore, we invite you to submit a revised version of the manuscript that addresses the points raised during the review process.

We look forward to receiving your revised manuscript.

Kind regards,

Vakhtang Tchantchaleishvili

Academic Editor

PLOS ONE

Journal Requirements:

2. In the ethics statement in the manuscript and in the online submission form, please provide additional information about the patient records used in your retrospective study. Specifically, please ensure that you have discussed whether all data were fully anonymized before you accessed them and/or whether the IRB or ethics committee waived the requirement for informed consent. If patients provided informed written consent to have data from their medical records used in research, please include this information.

Reviewers' comments:

Reviewer's Responses to Questions

**Comments to the Author**

Reviewer #1: This is an interesting hypothesis generating study investigating the effects of hemolysis in LVAD patients.

Overall, the hypothesis is clinical relevant and tested appropriately, the Methods are appropriate and the results are interesting. The sample size is small and this is a limitation. Otherwise, I would strongly encourage the authors to proof read their manuscript as there are several grammatical errors. Also, the authors need to provide more evidence about the definition of proximal tubulopathy that they used. The validity of the results depends of the accuracy and reproducibility of the definition of PT as a gold standard. Are there validation studies on non LVAD populations? Also, is there any reason that these criteria are reliable in LVAD patients? Also, it would be helpful if the authors provide any info about power spikes or need for device exchange in the long-term. Finally, please remove Tables and Figures from the main text and transfer them after the references.

Reviewer #2: I want to say thank you to the authors, I have read this manuscript with much interest and found it to be a very enjoyable read. This article points out the role of LDH in diagnosing this common comorbidity following LVAD implantation, as well as suggests the appropriate clinical cut off point that the clinicians should be aware of in susceptible patients. It also appropriately addresses many branching points of argument to be had regarding the findings. However I also agree that larger multi-centered cohort study should be warranted in the future to confirm the findings.

One thing I want to point out is that the numbers don't seem to match up, as authors initially claim 43 patients retrospectively studied however the data suggests there were only 33. Even though the authors do address the fact the 10 patients had to be excluded due to lack of data and their early mortality, I would make it clear in the body of text that in fact only 33 patients were accounted for. Aside from this small point I have no further recommendations to make.

Reviewer #3: Firstly, the authors should be congratulated for their work on studying the impact of sub-clinical hemolysis in patients implanted with Left ventricular assist devices and its association with proximal tubulopathy and renal function. However, there are certain issues about the study/manuscript which need to be addressed:

1. The authors in the discussion state "their study shows that in patients with Heartmate II device, chronic sub clinical hemolysis is frequent and associated with renal outcomes, especially functional and structural hallmarks of proximal tubular injury". They have however, not described structural changes of proximal tubulopathy in the entire manuscript. Since it is retrospective study with its inherent limitations, where the authors able to study/report structural changes of proximal tubulopathy in their LVAD patients (by light microscopy or EM).Obviously this would have required Renal biopsy and whether this was done in LVAD patients (on anticoagulation).

It would rather be their conclusion for proximal tubulopathy was made per renal functional assessment rather than structural changes.Hence, this conclusion needs to changed in the manuscript..

2.While it is known that outflow cannula angulation, pump speed with hemodynamic flow and sheer stress has effect on hemolysis and thrombogenecity, did the authors assess these effects of pump function (including pump speed) and cannula position in their patient subset. Was there any difference in PT (proximal tubulopathy) and no PT group in terms of pump function and outflow cannula position.

3. The authors have defined their criteria for proximal tubulopathy and one of them is proteinuria. Where they able to retrieve urine analysis report of these patients from their search? Where they able to differentiate proteinuria to be of glomerular versus tubular origin?

4. Please provide abbreviations for HO-1 (Heme oxygenase) on page 17,line 318

and MDRD (modification of diet in renal disease ) on page 14, line 235.

5. Please change all LDH units in the manuscript as IU/L (as in page 15 ,line 273 change LDH 600 UI/L to 600 IU/L).

6. For Figure 3, heading of ROC plot for LDH levels should be inserted;

with plot of AUC 0.83 (95% CI 0.68-0.98 and cut-off value of 600 IU/L) depicted in the figure.

Reviewer #4: The authors write on an interesting topic of subclinical hemolysis in MCS patients and its role on acute kidney injury and more long-term pathologies. Although an important topic and one that has not been heavily discussed in the literature, the authors fail to make substantial contributions to the topic due to issues with sample size, study design, and statistics.

The observation made by the authors that LDH is associated with PT in LVAD patients is valuable. Additionally the technical design of the study is sound with thorough clinical definitions and inclusion criteria for the various clinical subgroups that are used in the study (ie. hemolysis, PT). However there are concerns:

1- The power of the study is a limitation, so this claim must be taken in context. It is understood this is an inherent issue that can not be adjusted, but it does practically affect the findings.

2- The authors find their main conclusions by stratifying the cohort based on an outcome - PT. Although this is unavoidable in certain retrospective designs, in this case it appears a post-hoc sub-stratification of the cohort based on PT took place. Then additional analyses were done to assess for further differences between the groups. If this stratification was not done post-hoc, then the process of designing the study from hypothesis - to creation of stratifying variables, and predetermined outcomes needs to be better explained in the methods section. If this stratification was indeed done post-hoc this introduces significant potential for confounding in the final results - particularly as LDH is such a systemic biomarker affected by many pathophysiologic changes - much like renal injury.

3-The major concern with the paper focuses on use of the ROC curve and the claims associated. Although technically appropriate, the AUC analysis is not done in the most robust manner. As such, the broad claims by the authors that LDH can be used as a simple tool by physicians to raise alarm of kidney injury is questionable. Based on the limited information provided in the the methods section, there appears that there is no 'test' group with which the ROC curve could be analyzed. If the entire small cohort is stratified based on PT, and then differences in LDH are noticed between the groups, it is not ideal to then subsequently run an ROC curve analysis based on the same data to predict the same grouped outcomes. In a way, this method is just reaffirming the initial finding that there is a difference in LDH between the groups - not that it is in anyway predictive. To truly get the sensitivity and specificity of an LDH cutoff to predict an outcome, one needs to apply that finding to a uniquely different cohort. Ideally, the cohort should be partitioned where a subset of patients, not in the initial analysis, has the proposed LDH cutoff applied to their cases to observe if it is predictive of PT.

Reviewer #5: This article by Nattes et al presents data from a single center retrospective study of patients with stage D heart failure who underwent LVAD implantation, with emphasis on investigating the impact of LVAD-induced subclinical hemolysis on proximal tubulopathy (PT) and acute kidney injury (AKI) leading to chronic kidney disease in the LVAD population. The authors cite previous studies indicating that renal function improves transiently following LVAD implantation though subsequent decline in GFR occurs due to renal parenchymal damage, but also identify limitations of previous studies including the sparse of reports on the effects of subclinical hemolysis, identified based on increased LDH without overt pump thrombosis, on PT and AKI. The authors analyzed a total of 33 patients supported with an LVAD demonstrating that LVAD-related hemolysis was associated with PT which contributed to the development of AKI and subsequent CKD, concluding that reducing the incidence of hemolysis post LVAD implantation may thwart the development of CKD by decreasing the risk of PT in the LVAD population. To some extent, it remains unclear whether there the occurrence of CKD is explained by subclinical hemolysis, heart failure severity, comorbidities, and/or confounding.

The authors have attempted to answer these questions making use of retrospective data collection in a single center implanting Heartmate II LVADs between 2006 and 2017 ROC analysis showed that LDH threshold >600 UI/L was associated with a sensitivity of ~86% and specificity of ~85% for PT with an AUC of 0.83. PT definition based on general major and minor criteria determined by the authors is not well established and has not been previously validated. Using this definition, the authors found that Pt was associated with higher number of AKI events with the later being associated with increased incidence of CKD and death.

The paper is well written, the methods utilized by the authors are quite rigorous and the findings are fairly straightforward. The main limitations of this analysis are the retrospective analyses, reliance on unclear and less established criteria of PT, and lack of longitudinal data follow-up and other markers of hemolysis to confirm their results.

Some comments and suggestions for the authors' consideration:

- The authors stated in the abstract that 43 patients were analyzed while in the methods, 10 patients were included and only 33 patients were included in the study and finally analyzed. This should be corrected.

- The INTERMACS classification of the LVAD patients should be included. How many patients were bridged with temporary mechanical circulatory support devices (IABP, Impella, VA-ECMO)? Were there patients admitted with acute decompensated heart failure? Acute renal failure? Did any patients require aggressive diuresis prior to proceeding with LVAD surgery? Any required dialysis pre LVAD?

- Another missing information is regarding other hemolysis markers such as plasma-free hemoglobin and hematuria. The statement by the authors that plasma-free hemoglobin, for instance, has not been performed to have real-life reproducible evaluation is unclear to the reviewer.

- A major limitation of this study is the inclusion of criteria that have not previously validated as specific for PT. How did the authors select these criteria? Were there any previous studies using these criteria of PT? The reviewer can find other causes proteinuria, glycosuria without hyperglycemia, hypokalemia <3.5, etc.. in other clinical scenarios that do not necessarily indicate PT.

- What was the HBA1c values in this cohort and among diabetic patients? A sensitivity analysis excluding patients with diabetes may be required.

- Several baseline characteristics are missing and may confound the results. These include, INTERMACS class, BTT vs. BT indication, blood pressure, resternotomy, albumin and bilirubin levels in plasma, ICD implantation, and hemoglobin levels at baseline. These characteristics need to be compared between the Pt and non-Pt groups as well with p values provided.

- Besides GFR and creatinine values at baseline, urine protein level at baseline is a significant predictor for AKI post LVAD. Have the authors included this in the analysis?

- In addition to references #5 and #11, the authors may also cite the paper “Asleh et al. Predictors and Outcomes of Renal Replacement Therapy After Left Ventricular Assist Device Implantation. Mayo Clin Proc. 2019 Jun;94(6):1003-1014.”

- What was the length of follow-up?

- Have the authors adjusted for potential confounders for mortality among LVAD patients?

- In Results, line 161: “Two patients (6.0%) had proteinuria” – How much protein in urine? In which group (PT vs. non-PT)?

- In Results, line 168, “6 underwent acute hemodialysis… after LVADs implantation” What was the baseline creatinine and urine protein levels in this group? How many developed PT based on the authors definition? What was there outcome? This should be clarified.

- It appears that PT was associated with AKI episodes and AKI was associated with CKD. The authors concluded that PT leads to CKD. This is unclear from the analysis. Was there a direct association between PT and CKD incidence by logistic regression analysis?

- Based on the CKD results, it seems that GFR and creatinine were similar between the PT and non-PT groups at end of follow-up. This should be discussed and clarified as it contradicts the study conclusions that PT may increase the incidence of CKD.

- It appears based on the ROC analysis that most patients had LDH average of 500-600. Was there a linear correlation with PT/AKI. In other words, did patients with higher values have higher probability of PT that can support causality?

Minor comments:

- There several typos that require extensive grammar and language revision: For example: In the abstract (methods, line 37): PT groups were defined according proteinuria (missing “to”). In the abstract (lines 40-41): the word “respectively” should be at the end of the sentence not in the beginning. In the introduction (line 82): accelerated degradation of GFR – degradation is inappropriate here and maybe replaced with “decline in”

- In Results, second paragraph (lines 200-202) is unclear and needs to be clarified/rewritten.

- In Results, lines 110-111 – “Alternative etiologies for hemolysis…” Which alternative etiologies were excluded? Have any patients had pump thrombosis or pump exchange? What were the INR values at the time of LDH measurements? Which medical regimen was used in this population (antiplatelet doses and INR target values)?

- In the Discussion (lines 307-310) appears less relevant to the study and can be omitted.

- Please define HO-1 in line 318.

- In the conclusion (lines 344-345)- The authors did not demonstrate that PT directly resulted in renal outcomes and death. Therefore, this statement should be tuned down.

6. PLOS authors have the option to publish the peer review history of their article (what does this mean?). If published, this will include your full peer review and any attached files.

---

## [Author Response · Author response to Decision Letter 0]

2 Oct 2020

Dear Editor-In-Chief,

We wish to address our warmest thanks to the Editor and Referees of Plos One for providing us with the opportunity to revise our manuscript. We have carefully addressed each of the points raised by the reviewers, provided a point-by-point answer and modified the manuscript accordingly. As a result, we believe our manuscript has been significantly improved thanks to these constructive comments. Should there be any further queries we would be more than happy to answer them. 

On behalf of all authors, Tristan de Nattes

Reviewer #1: This is an interesting hypothesis generating study investigating the effects of hemolysis in LVAD patients. Overall, the hypothesis is clinical relevant and tested appropriately, the Methods are appropriate and the results are interesting. The sample size is small and this is a limitation. 

Otherwise, I would strongly encourage the authors to proofread their manuscript as there are several grammatical errors. 

We thank you for the time you took to evaluate our manuscript and for your positive feedback on our findings. We apologize for the grammatical errors. The revised manuscript has been proofread by a native language speaker. 

Also, the authors need to provide more evidence about the definition of proximal tubulopathy that they used. The validity of the results depends of the accuracy and reproducibility of the definition of PT as a gold standard. Are there validation studies on non LVAD populations? Also, is there any reason that these criteria are reliable in LVAD patients? 

To date, there is no validated score for the diagnosis of Proximal Tubulopathy. Owing to its physiological function (1), the typical features of PT are low-molecular weight proteinuria (2), glycosuria and phosphaturia. Besides these signs, several indicators of Proximal Tubular Dysfunction are validated, mainly in management guidelines of patients infected by HIV who are particularly exposed to drug-induced PT, especially secondary to Tenofovir (3)(4). Owing to that, red flags in the follow-up of these patients are: hypokaliemia, low serum bicarbonate, hypophosphatemia with abnormal fractional excretion of phosphate (5), abnormal fractional excretion of uric acid, glycosuria (6), aminoacidura and urinary B2 microglobulin (7). These signs are validated in other diseases as Lowe syndrome (8,9), drug-toxicities (10), and have been recently used as a cornerstone for the diagnosis of PT in the scope of COVID (11). 

Due to its retrospective design, it has been necessary to choose, among those listed, biological elements for which the laboratory techniques did not change across years and which are widely performed in daily monitoring of LVAD patients. Consequently, we selected proteinuria and glycosuria as major criteria considering that these signs are more specific of tubular alterations in this setting. Owing to the fact that hypokalemia, serum phosphate and aseptic leukocyturia may reflect other issues in patients with LVAD (e.g diarrhea, malnutrition, treated urinary tract infection…), we made the choice to consider them as minor criteria. At least, specific signs as aminoaciduria or urinary B2 microglobulin were not routinely performed in our unit and were consequently excluded from the analysis. Finally, biological data were collected only in stable periods and patients were classified in the PT group only when two samples several weeks apart revealed the same findings. 

Although there is no validated score to formally diagnose PT, we believe that these elements are clinically relevant and correspond to usual diagnostic criteria.

We added this in the “discussion” section of the revised manuscript, page 15

Also, it would be helpful if the authors provide any info about power spikes or need for device exchange in the long-term. 

Four patients required a pump exchange: two because of pump dysfunction, and one because of cannula position. These three patients were classified in “no PT” group. The fourth exchange was required because of pump thrombosis. Considering that the interpretation of pump power elevations is not clear, we did not include this data in our analyses (12). We revised the manuscript accordingly page 11, line 197:

“Six patients had a LVAD thrombosis: 4 patients (15.4%) in the “no PT” group and 2 patients (28.6%) in the “PT” group. All patients were classified in their PT group before occurrence of LVAD thrombosis, and all patients except one died after the pump thrombosis. Three other patients required a pump exchange, all classified in the “no PT” group”

Finally, please remove Tables and Figures from the main text and transfer them after the references.

As requested by Plos One submission guideline, figure captions are inserted in the text of the manuscript, and tables appear directly after the paragraph in which they are cited.

Reviewer #2: I want to say thank you to the authors, I have read this manuscript with much interest and found it to be a very enjoyable read. This article points out the role of LDH in diagnosing this common comorbidity following LVAD implantation, as well as suggests the appropriate clinical cut off point that the clinicians should be aware of in susceptible patients. It also appropriately addresses many branching points of argument to be had regarding the findings. However I also agree that larger multi-centered cohort study should be warranted in the future to confirm the findings.

We kindly thank the reviewer for his appreciation.

One thing I want to point out is that the numbers don't seem to match up, as authors initially claim 43 patients retrospectively studied however the data suggests there were only 33. Even though the authors do address the fact the 10 patients had to be excluded due to lack of data and their early mortality, I would make it clear in the body of text that in fact only 33 patients were accounted for. Aside from this small point I have no further recommendations to make.

As the reviewer mentioned, 43 patients were implanted with a Heartmate II device and 10 were excluded from the analysis due to the lack of data. We clarified this point at the beginning of the “results” section.

“Forty-three patients were implanted, four patients were excluded because their survival was less than 60 days and six patients were excluded from the Proximal Tubulopathy analysis because of insufficient biological data. Clinical characteristics of the remaining 33 patients are summarized in table 2”

Reviewer #3: Firstly, the authors should be congratulated for their work on studying the impact of sub-clinical hemolysis in patients implanted with Left ventricular assist devices and its association with proximal tubulopathy and renal function. However, there are certain issues about the study/manuscript which need to be addressed:

We thank you for your appreciation and are pleased by your enthusiasm.

1. The authors in the discussion state "their study shows that in patients with Heartmate II device, chronic sub clinical hemolysis is frequent and associated with renal outcomes, especially functional and structural hallmarks of proximal tubular injury". They have however, not described structural changes of proximal tubulopathy in the entire manuscript. Since it is retrospective study with its inherent limitations, where the authors able to study/report structural changes of proximal tubulopathy in their LVAD patients (by light microscopy or EM). Obviously this would have required Renal biopsy and whether this was done in LVAD patients (on anticoagulation).

It would rather be their conclusion for proximal tubulopathy was made per renal functional assessment rather than structural changes. Hence, this conclusion needs to changed in the manuscript.

Thank you for this comment. Indeed, no kidney biopsy was performed in our unit due to curative anticoagulation. Only one histology was available in a single patient after autopsia performed for an unexplained sudden death 695 days after HeartMate II implantation, with an average LDH value of 700 UI/L. Kidney findings revealed interstitial fibrosis, Perls’ Prussian blue staining showed abundant proximal tubular intracytoplasmic and interstitial iron deposits, the glomeruli and vessels were normal. Due to space constraints and considering that there was only one single case, we chose not to detail this pathological finding in our manuscript, but could of course include it if requested. Please find here a picture of the histology. 

Figure: Renal hemosiderosis in a 57-years old patient with chronic intravascular hemolysis due to an Heartmate II device. Abundant proximal tubular intracytoplasmic and interstitial iron deposits (Perls’ Prussian blue staining 20x).

We suppressed the mention to structural changes from our manuscript (page 14 line 249; page 16 line 291). 

2.While it is known that outflow cannula angulation, pump speed with hemodynamic flow and sheer stress has effect on hemolysis and thrombogenecity, did the authors assess these effects of pump function (including pump speed) and cannula position in their patient subset. Was there any difference in PT (proximal tubulopathy) and no PT group in terms of pump function and outflow cannula position.

We thank the reviewer for this relevant comment. In agreement with these concerns, our practices include a precise perioperative control of cannula position by US-echography in order to obtain the best position. Moreover, in the post-operative period, patients are regularly evaluated with US-echography to obtain the most appropriate pump-speed providing optimal pump outflow. In the same time, we control the pump data files to detect any events of suction events. 

Although these parameters have an influence on shear stress and thrombogenicity as we have discussed in the manuscript (page 18, line 352), we believe that it is unlikely that they may have a direct influence on proximal tubular cells, which is not supported by literature. Consequently, a possible difference would, in our opinion, rather be responsible for a greater hemolysis and in turn for a renal toxicity. In order not to add confusion the main message, we believe that this information can be omitted.

3. The authors have defined their criteria for proximal tubulopathy and one of them is proteinuria. Where they able to retrieve urine analysis report of these patients from their search? Where they able to differentiate proteinuria to be of glomerular versus tubular origin?

In our unit, urine protein electrophoresis in not systematically performed. However, urine albumin was available in more than the half of patients. In these patients, urine albumin excretion was not significantly increased, allowing us to exclude proteinuria from glomerular origin. In other patients, we considered usual hallmarks of tubular injury, i.e a low abundance around 1 g/day as compared to glomerular proteinuria which is supposed to be higher than 2 g/day (2). Moreover, we paid particular attention to hematuria, which is a classical sign of glomerular lesions, and therefore associated with glomerular proteinuria. As detailed in the “results” section (page 11, line 18), none of the patients had hematuria, which is another argument for a proteinuria from tubular origin. 

We added this discussion in the “discussion” section of our manuscript, page 15, as below:

“In order to differentiate proteinuria of glomerular versus tubular origin, we considered usual hallmarks of tubular injury, i.e a low abundance around 1 g/day as compared to glomerular proteinuria which is supposed to be higher than 2 g/day (16). Moreover, we paid particular attention to hematuria, which is a classical sign of glomerular lesions, and therefore associated with glomerular proteinuria. In our study, no patient had hematuria, which is another argument for a proteinuria from tubular origin. Owing to the fact that hypokalemia, serum phosphate and aseptic leukocyturia may reflect other issues (e.g diarrhea, malnutrition, treated urinary tract infection…), we made the choice to consider them as minor criteria.”

4. Please provide abbreviations for HO-1 (Heme oxygenase) on page 17,line 318

and MDRD (modification of diet in renal disease ) on page 14, line 235.

5. Please change all LDH units in the manuscript as IU/L (as in page 15 ,line 273 change LDH 600 UI/L to 600 IU/L).

We apologize for these mistakes. We corrected them in the manuscript. 

6. For Figure 3, heading of ROC plot for LDH levels should be inserted;

with plot of AUC 0.83 (95% CI 0.68-0.98 and cut-off value of 600 IU/L) depicted in the figure.

Thank you for this advice. We corrected it accordingly.

Reviewer #4: The authors write on an interesting topic of subclinical hemolysis in MCS patients and its role on acute kidney injury and more long-term pathologies. Although an important topic and one that has not been heavily discussed in the literature, the authors fail to make substantial contributions to the topic due to issues with sample size, study design, and statistics.

The observation made by the authors that LDH is associated with PT in LVAD patients is valuable. Additionally the technical design of the study is sound with thorough clinical definitions and inclusion criteria for the various clinical subgroups that are used in the study (ie. hemolysis, PT). However there are concerns:

1- The power of the study is a limitation, so this claim must be taken in context. It is understood this is an inherent issue that can not be adjusted, but it does practically affect the findings.

We thank you for your proof reading and your kind appreciation. 

We are in full agreement with this concern. We better highlighted these limitations in our manuscript and hope that this study will be of interest to centers with a larger cohort and therefore be the first step in further investigations. Indeed, the complexity of the renal evolution of these patients, which is illustrated by contradictory results as mentioned in the “introduction” section, raises the major interest of exploratory studies. We adjusted the “discussion” and “conclusion” section accordingly 

“Despite the sample size, this study shows that in patients with Heartmate II device, chronic subclinical hemolysis is frequent and associated with renal outcomes, especially functional and structural hallmarks of proximal tubular injury” (page 14, line 247)

 “Therefore, it should be investigated in a larger confirmation cohort” (page 19, line 353)

2- The authors find their main conclusions by stratifying the cohort based on an outcome - PT. Although this is unavoidable in certain retrospective designs, in this case it appears a post-hoc sub-stratification of the cohort based on PT took place. Then additional analyses were done to assess for further differences between the groups. If this stratification was not done post-hoc, then the process of designing the study from hypothesis - to creation of stratifying variables, and predetermined outcomes needs to be better explained in the methods section. If this stratification was indeed done post-hoc this introduces significant potential for confounding in the final results - particularly as LDH is such a systemic biomarker affected by many pathophysiologic changes - much like renal injury.

3-The major concern with the paper focuses on use of the ROC curve and the claims associated. Although technically appropriate, the AUC analysis is not done in the most robust manner. As such, the broad claims by the authors that LDH can be used as a simple tool by physicians to raise alarm of kidney injury is questionable. Based on the limited information provided in the the methods section, there appears that there is no 'test' group with which the ROC curve could be analyzed. If the entire small cohort is stratified based on PT, and then differences in LDH are noticed between the groups, it is not ideal to then subsequently run an ROC curve analysis based on the same data to predict the same grouped outcomes. In a way, this method is just reaffirming the initial finding that there is a difference in LDH between the groups - not that it is in anyway predictive. To truly get the sensitivity and specificity of an LDH cutoff to predict an outcome, one needs to apply that finding to a uniquely different cohort. Ideally, the cohort should be partitioned where a subset of patients, not in the initial analysis, has the proposed LDH cutoff applied to their cases to observe if it is predictive of PT.

We agree with these comments and have better highlighted these limitations in the revised manuscript. We added these informations page 19, line 353.

“Since the study population was stratified post-hoc based on the PT score, potential confounding cannot be excluded. In addition, a validation would theoretically have been required to ascertain the ROC analysis performed.”

Reviewer #5: This article by Nattes et al presents data from a single center retrospective study of patients with stage D heart failure who underwent LVAD implantation, with emphasis on investigating the impact of LVAD-induced subclinical hemolysis on proximal tubulopathy (PT) and acute kidney injury (AKI) leading to chronic kidney disease in the LVAD population. The authors cite previous studies indicating that renal function improves transiently following LVAD implantation though subsequent decline in GFR occurs due to renal parenchymal damage, but also identify limitations of previous studies including the sparse of reports on the effects of subclinical hemolysis, identified based on increased LDH without overt pump thrombosis, on PT and AKI. The authors analyzed a total of 33 patients supported with an LVAD demonstrating that LVAD-related hemolysis was associated with PT which contributed to the development of AKI and subsequent CKD, concluding that reducing the incidence of hemolysis post LVAD implantation may thwart the development of CKD by decreasing the risk of PT in the LVAD population. To some extent, it remains unclear whether there the occurrence of CKD is explained by subclinical hemolysis, heart failure severity, comorbidities, and/or confounding.

The authors have attempted to answer these questions making use of retrospective data collection in a single center implanting Heartmate II LVADs between 2006 and 2017 ROC analysis showed that LDH threshold >600 UI/L was associated with a sensitivity of ~86% and specificity of ~85% for PT with an AUC of 0.83. PT definition based on general major and minor criteria determined by the authors is not well established and has not been previously validated. Using this definition, the authors found that Pt was associated with higher number of AKI events with the later being associated with increased incidence of CKD and death.

The paper is well written, the methods utilized by the authors are quite rigorous and the findings are fairly straightforward. The main limitations of this analysis are the retrospective analyses, reliance on unclear and less established criteria of PT, and lack of longitudinal data follow-up and other markers of hemolysis to confirm their results.

We thank the reviewer for his analysis and the quality of his report.

Some comments and suggestions for the authors' consideration:

- The authors stated in the abstract that 43 patients were analyzed while in the methods, 10 patients were included and only 33 patients were included in the study and finally analyzed. This should be corrected.

We apologize for this unclear point. Forty-three patients were implanted, in which 10 were excluded from the analysis due to insufficient data. We corrected the abstract accordingly.

“Methods: Thirty-three patients implanted with a Heartmate II LVAD (Abbott, Inc, Chicago IL) were retrospectively studied.”

- The INTERMACS classification of the LVAD patients should be included.

- What was the HBA1c values in this cohort and among diabetic patients? A sensitivity analysis excluding patients with diabetes may be required.

- Several baseline characteristics are missing and may confound the results. These include, INTERMACS class, BTT vs. BT indication, blood pressure, resternotomy, albumin and bilirubin levels in plasma, ICD implantation, and hemoglobin levels at baseline. These characteristics need to be compared between the Pt and non-Pt groups as well with p values provided.

-How many patients were bridged with temporary mechanical circulatory support devices (IABP, Impella, VA-ECMO)? Were there patients admitted with acute decompensated heart failure?

We thank the reviewer for these comments and advices. To improve clarity, we have merged his comments above. All these data have been added in table 2 with p values as indicated below. Due to the absence of statistically significant differences between groups according to these characteristics, no other analyses were performed.

 PT, n=7 (21.1%) No PT, n=26 (78.8) p

Baseline characteristics 

Age (year) 71 [61-72] 62 [57-66] 0.07

Diabetes, n (%) 2 (28.6) 5 (19.2) 0.6

HbA1c (%) 7.8 [7.2-8.4] 7.2 [7.0-8.0] 0.5

Hypertension, n (%) 5 (71.4) 18 (69.2) 0.9

Plasma creatinine at the implantation (µmol/L) 113 [86-137] 106 [72-150] 0.6

eGFR (mL/min/1.73m2) 57 [48-83] 65 [43-96] 0.6

Biological data 

• Hemoglobin (g/dL)

• Total bilirubin (µmol/L)

• Albumin (g/dL) 

10.9 [9.9-11.7]

19.0 [13.0-44.0]

32.0 [30.7-34.0] 

11.8 [9.5-13.0]

19.0 [12.8-31.3]

29.3 [25.5-32.0] 

0.3

0.9

0.2

Blood pressure at the implantation (mmHg)

• Systolic

• Diastolic 

110 [84-115]

70 [50-77] 

100 [88-120]

70 [55-90] 

0.9

0.5

Cause of heart failure, n (%)

• Ischemic

• Dilative

• Myocarditis 

4 (57.1)

3 (42.9) 

0 

17 (65.4)

8 (30.8)

1 (3.8)

0.7

0.5

INTERMACS classification, n (%)

• 1

• 2

• 3

• 4

• 5-7 

2 (28.6)

2 (28.6)

1 (14.3)

2 (28.6)

0 

5 (19.2)

13 (50.0)

5 (19.2)

2 (7.7)

1 (3.8) 0.5

Indication

• Destination therapy, n (%)

• Bridge to transplantation, n (%) 

3 (42.9)

4 (57.1) 

5 (19.2)

21 (80.8) 0.2

Temporary mechanical circulatory support, n (%)

• Impella

• ECMO 

0

6 (85.7) 

9 (34.6)

13 (50.0) 0.05

Resternotomy, n (%) 1 (14.3) 4 (15.4) 0.9

ICD implantation, n (%) 5 (71.4) 16 (61.5) 0.6

Follow-up characteristics 

HMII support duration (days) 511 [332-1091] 715 [359-1186] 0.9

LDH 688 [643-703] 356 [320-494] 0.006

Acquired von Willebrand syndrome, n (%) 3 (42.3) 14 (53.8) 0.6

Transfusion (n) 21 [12-36] 29 [16-54] 0.5

Device or driveline infection 6 (85.7) 20 (76.9) 0.6

AKI (n of episode/year) 4.3 [2.5-5.0] 1.6 [0.4-3.7] 0.05

Plasma creatinine at the end of follow-up 141 [62-152] 100 [87-149] 0.6

eGFR (mL/min/1.73m2) 46 [42-118] 64 [43-80] 0.6

End of follow-up, n (%)

• Transplanted

• Dead

o Destination therapy

• Alive with HMII

o Destination therapy 

3 (42.9)

4 (57.1)

 3 (75.0)

15 (57.7)

6 (23.1)

4 (66.7)

5 (19.2)

1 (20.0) 

0.7

0.2

Table 1 : characteristics of patients according to their Proximal Tubulopathy (PT) group. Values are medians [25th-75th percentile]. ECMO: Extracorporeal Membrane Oxygenation. ICD: Implantable Cardioverter Defibrillators. HMII: Heartmate II. LDH: Lactate Dehydrogenase. AKI: Acute Kidney Injury. eGFR: estimated Glomerular Filtration Rate. 

[Were there patients admitted with] acute renal failure? Any required dialysis pre LVAD? Did any patients require aggressive diuresis prior to proceeding with LVAD surgery?

As mentioned in the “introduction” section, the evaluation of renal function in the scope of LVAD is subject to many biases. Indeed, a significant proportion of our patients were admitted in ICU with instable and severe acute heart failure requiring mechanical assist device. In this context, kidney function is likely to be impaired through different pathways: hypoperfusion, acute tubular necrosis, drugs toxicities. Using last stable plasma creatinine available before hospitalization for Heartmate II implantation as an estimator for baseline renal function, 26 patients (78.8 %) presented an increase in plasma creatinine > 26.5 µmol/L or > 1.5 fold baseline (13). None required hemodialysis before LVAD implantation, while all patients received high-dose intravenous diuretics.

We added this information in the “results” section, page 10 line 181:

“Twenty-six patients (78.8%) had AKI between hospitalization and LVAD implantation, without requiring hemodialysis. Nineteen patients (57.6%) had AKI the first month after Heartmate II implantation. Six patients (18.2%) had AKI between the first month and hospital discharge.”

- Another missing information is regarding other hemolysis markers such as plasma-free hemoglobin and hematuria. The statement by the authors that plasma-free hemoglobin, for instance, has not been performed to have real-life reproducible evaluation is unclear to the reviewer.

We thank the reviewer for this comment.

Renal toxicity of hemolysis is related to several mechanisms, mainly due to the heme. To avoid damage caused by hemolysis, plasma-free hemoglobin is scavenged by haptoglobin (14). When this system is saturated, free heme is released and scavenged by hemopexin. Then, heme is metabolized into biliverdin by the Heme Oxygenase (HO) system, which generates CO and releases iron. HO-1 expression is induced in tubules by oxidative stress, and by recognition of haemoglobin/haptoglobin and heme/hemopexin complexes. HO-2 is constitutively and abundantly present in the thick ascending limb, distal tubule, and preglomerular vasculature (15). Localization of these proteins explains the sensitivity of tubular proximal cells to hemolysis (16). Considering these biological steps, the presence of plasma-free hemoglobin and/or hemoglobinuria (rather than hematuria) correspond to an intense hemolysis exceeding adaptation capacity, while LDH value with simultaneous haptoglobin < 0.1 g/L offers to the clinician a way to evaluate and monitor low ongoing subclinical hemolysis. Methods for measuring free hemoglobin (chromogenic, spectrophotometric or immunonephelometric assays) have been modified across years, preventing their use for a retrospective long-term monitoring (17). More, the actual Gold-standard using the Harboe method is not fully automatised, leading to a lack of reproducibility, an increased cost, and a limited availability of this test in other centres (18). For all these reasons, and in order to facilitate the diffusion of our results, we chose to monitor the LDH value, which is based on widely and easily validated methods.

In order to clarify our manuscript, we suppressed this mention from the “method” section page 7 line 128: We did not consider plasma-free hemoglobin in order to have a real-life reproducible evaluation. 

- A major limitation of this study is the inclusion of criteria that have not previously validated as specific for PT. How did the authors select these criteria? Were there any previous studies using these criteria of PT? The reviewer can find other causes proteinuria, glycosuria without hyperglycemia, hypokalemia <3.5, etc.. in other clinical scenarios that do not necessarily indicate PT.

We thank the reviewer for his comment, which was also raised by reviewer #1. Please find our answer below. 

To date, there is no validated score for the diagnosis of Proximal Tubulopathy. Owing to its physiological function (1), the typical features of PT are low-molecular weight proteinuria (2), glycosuria and phosphaturia. Besides these signs, several indicators of Proximal Tubular Dysfunction are validated, mainly in management guidelines of patients infected by HIV who are particularly exposed to drug-induced PT, especially secondary to Tenofovir (3)(4). Owing to that, red flags in the follow-up of these patients are: hypokaliemia, low serum bicarbonate, hypophosphatemia with abnormal fractional excretion of phosphate (5), abnormal fractional excretion of uric acid, glycosuria (6), aminoacidura and urinary B2 microglobulin (7). These signs are validated in other diseases as Lowe syndrome (8,9), drug-toxicities (10), and have been recently used as a cornerstone for the diagnosis of PT in the scope of COVID (11). 

Due to its retrospective design, it has been necessary to choose, among those listed, biological elements for which the laboratory techniques did not change across years and which are widely performed in daily monitoring of LVAD patients. Consequently, we selected proteinuria and glycosuria as major criteria considering that these signs are more specific of tubular alterations in this setting. Owing to the fact that hypokalemia, serum phosphate and aseptic leukocyturia may reflect other issues in patients with LVAD (e.g diarrhea, malnutrition, treated urinary tract infection…), we made the choice to consider them as minor criteria. At least, specific signs as aminoaciduria or urinary B2 microglobulin were not routinely performed in our unit and were consequently excluded from the analysis. Finally, biological data were collected only in stable periods and patients were classified in the PT group only when two samples several weeks apart revealed the same findings. Although there is no validated score to formally diagnose PT, we believe that these elements are clinically relevant and correspond to usual diagnostic criteria.

We added this in the “discussion” section of the revised manuscript, page 15

- In addition to references #5 and #11, the authors may also cite the paper “Asleh et al. Predictors and Outcomes of Renal Replacement Therapy After Left Ventricular Assist Device Implantation. Mayo Clin Proc. 2019 Jun;94(6):1003-1014.”

We thank the reviewer for this advice. We added the reference in the indroduction page 6 line 95.

It has also been shown by Muslem et al that pre-operative proteinuria before LVADs implantation is associated with mortality and hemodialysis at one year (19). More, the need of hemodialysis after LVADs implantation is associated with an increased risk of mortality, underlying the impact of renal function in LVADs patients survivial (12).

- What was the length of follow-up?

Follow-up was stopped when LVADs were definitely explanted: heart transplantation or death. In the PT the median follow-up was 511 days [332 – 1091] versus 715 days [359 – 1186] in the no PT group, p=0.9. This information has been added in table 2. 

- Have the authors adjusted for potential confounders for mortality among LVAD patients?

Owing to the size of the study sample, adjustment for potential confounders for mortality was not performed. 

- In Results, line 161: “Two patients (6.0%) had proteinuria” – How much protein in urine? In which group (PT vs. non-PT)?

- Besides GFR and creatinine values at baseline, urine protein level at baseline is a significant predictor for AKI post LVAD. Have the authors included this in the analysis?

Two patients (6.0%) had proteinuria: one patient in each group, without difference of proteinuria/creatininuria ratio which was 0.6 g/g for both. We added this information in the manuscript page 10 line 173. Because of the small number of patients, we did not perform statistical analysis based on this baseline characteristic.

“Two patients (6.0%) had proteinuria at baseline: one patient in each group (proteinuria/creatininuria ratio was 0.6 g/g for both of them)”

- In Results, line 168, “6 underwent acute hemodialysis… after LVADs implantation” What was the baseline creatinine and urine protein levels in this group? How many developed PT based on the authors definition? What was there outcome? This should be clarified.

Thank you for this relevant interrogation. Please find here a table in which main characteristics of patients according to the need of hemodialysis after LVADs implantation are summarized.

Characteristics Hemodialysis after LVADs implantation

N=7 No need for hemodialysis

N=26 p

Age (year) 65.0 [59.0 – 71.0] 60.0 [56.5 – 67.5] 0.35

Baseline creatinine, µmol/L 73 [60 – 127] 112 [81 – 152] 0.11

Baseline proteinuria, g/g 0.6 0.6 

Indication, n (%)

• Bridge to transplantation

• Destination 

4 (57.1)

3 (42.9) 

21 (80.8)

5 (19.2) 0.32

Cardiopathy, n (%)

• Ischemic

• Dilative

• Myocarditis 

4 (57.1)

3 (42.9)

0 

17 (65.4)

8 (30.8)

1 (3.8) 0.59

INTERMACS classification, n (%)

• 1

• 2

• 3

• 4

• 5-7 

1 (14.3)

3 (42.9)

3 (42.9)

0

0 

6 (23.1)

12 (46.2)

3 (11.5)

4 (15.4)

1 (3.8) 0.34

Temporary mechanical circulatory support, n (%)

• ECMO

• Impella

• None 

4 (57.1)

2 (26.6)

1 (14.3) 

14 (53.8)

8 (30.8)

4 (15.4) 0.99

AKI (n of episode/year) 4.6 [2.8 – 5.3] 2.1 [0.4 – 3.7] 0.06

Plasma creatinine at the end of follow-up, µmol/L 125 [60 – 150] 101 [87 – 150] 0.66

LDH, IU/L 361 [335 – 709] 385 [334 – 689] 0.94

Proximal tubulopathy, n (%)

• Yes

• No 

3 (42.9)

4 (57.1) 

3 (11.5)

23 (88.5) 0.09

Table 2: characteristics of patients according to the need of acute hemodialysis after LVADs implantation. Values are medians [25th-75th percentile]. ECMO: Extracorporeal Membrane Oxygenation. LDH: Lactate Dehydrogenase. AKI: Acute Kidney Injury. 

As reported in this table, there was no difference at baseline between groups defined by the need of hemodialysis after LVADs implantation. These finding differ from other studies, probably because of the low number of patients (20). The main points here are that acute hemodialysis does not impact the mean LDH value (361 versus 385 IU/L) neither the presence of PT (p = 0.09). Albeit the absence of statistically difference, the fact that patients who required acute hemodialysis have more episodes of acute kidney injury is in line with literature, reflecting the maladaptive repair following severe kidney aggression (21). 

Due to space constraints and in order to prioritize the primary end-points, we propose to add this table in supplementary file, and we modified our manuscript accordingly:

“Nineteen patients (57.6%) had AKI the first month after Heartmate II implantation. Six patients (18.2%) had AKI between the first month and hospital discharge. Among these patients, 7 underwent acute hemodialysis. There was no difference at baseline between patients who required acute hemodialysis and others. More, the need of acute hemodialysis after LVADs implantation did not influence the presence or absence of PT (supplementary table 1).”

- It appears that PT was associated with AKI episodes and AKI was associated with CKD. The authors concluded that PT leads to CKD. This is unclear from the analysis. Was there a direct association between PT and CKD incidence by logistic regression analysis?

This analyze was not performed due to the risk of bias. The point we highlighted in the manuscript is that chronic subclinical hemolysis is associated with PT and that patients with PT had an increased susceptibility to AKI. 

The last analysis concerning the association between the number of AKI and the occurrence of CKD was performed regardless of the PT statue. 

To avoid misinterpretation, we suppressed this statement from the conclusion.

- Based on the CKD results, it seems that GFR and creatinine were similar between the PT and non-PT groups at end of follow-up. This should be discussed and clarified as it contradicts the study conclusions that PT may increase the incidence of CKD.

Renal function at the end of the follow-up was not statistically different between the PT and no-PT group (table 3: plasma creatinine = 141 µmol/L versus 100 µmol/L, p=0.6, eGFR = 46 mL/min/1.73m2 versus 64 mL/min/1.72m2, p=0.6). This is probably related to the low number of patients. Analyzes concerning CKD were performed regardless of the PT status. We made this clear in the results section page 14 line 249. We also modified the discussion section, page 15 line 263:

“Third, patients with LVADs-related PT present an increased occurrence of AKI. Finally, regardless of the presence of PT, patients who had the more AKI episodes had the worst renal function at the end of the follow up”

And page 17 line 303:

“We found an association between PT and the number of AKI, suggesting that chronic proximal tubular functional alterations can impair kidney adaptability. Although we did not identify a direct association between PT and CKD, which could be explained by the low number of patients, patients with worst renal function at the end of follow-up had more episodes of AKI than patients with normal or mildly impaired renal function.”

- It appears based on the ROC analysis that most patients had LDH average of 500-600. Was there a linear correlation with PT/AKI. In other words, did patients with higher values have higher probability of PT that can support causality?

Indeed, patients with high values of LDH have a high probability of PT. However, owing to the categorical nature of the PT and AKI variables, linear analyses were not performed.

Minor comments:

- There several typos that require extensive grammar and language revision: For example: In the abstract (methods, line 37): PT groups were defined according proteinuria (missing “to”). In the abstract (lines 40-41): the word “respectively” should be at the end of the sentence not in the beginning. In the introduction (line 82): accelerated degradation of GFR – degradation is inappropriate here and maybe replaced with “decline in”

- Please define HO-1 in line 318.

We apologize for these mistakes. We corrected the manuscript accordingly. 

- In Results, second paragraph (lines 200-202) is unclear and needs to be clarified/rewritten.

We corrected this paragraph as follows:

“All patients had received blood transfusions: 21 [12-36] transfusions per patient in the “PT” group and 29 [16-54] transfusions per patient in the “no PT” group during the entire follow-up period, p=0.5.”

- In Results, lines 110-111 – “Alternative etiologies for hemolysis…” Which alternative etiologies were excluded? Have any patients had pump thrombosis or pump exchange? What were the INR values at the time of LDH measurements? Which medical regimen was used in this population (antiplatelet doses and INR target values)?

All the patients included in this study underwent extensive exploration in order to diagnose another etiology for their hemolytic anemia. There were no hereditary disorders (spherocytosis, sickle cell anemia, G6PD deficiency), Coombs’ tests were negative in all cases, and hemolytic anemia could not be explained by drugs’ toxicity, infectious agent or chronic autoimmune disease. We corrected the manuscript accordingly (page 7, line 111):

By retrospective review, alternative etiologies for hemolysis were excluded: There were no hereditary disorders (spherocytosis, sickle cell anemia, G6PD deficiency), Coombs’ tests were negative in all cases, and hemolytic anemia could not be explained by drugs’ toxicity, infectious agent or chronic autoimmune disease.

As mentioned in the “results” section, 4 patients (15.4%) in the “no PT” group had pump thrombosis versus 2 patients (28.6%) in the “PT” group (28.6%). Importantly, all patients were classified in their PT group before thrombosis. All patients except one died after the pump thrombosis. Three other patients had a pump exchanges, two because of pump dysfunction, and one because of cannula position. These three patients were classified in “no PT” group. We added this data page 11, line 197:

“Six patients had a LVAD thrombosis: 4 patients (15.4%) in the “no PT” group and 2 patients (28.6%) in the “PT” group. All patients were classified in their PT group before occurrence of LVAD thrombosis, and all patients except one died after the pump thrombosis. Three other patients required a pump exchanges, all classified in the “no PT” group”

In our unit, all patients were treated by anticoagulation therapy with a therapeutic INR range between 2.0 and 2.5. As previously published, we do not use antiplatelet therapy in order to reduce major bleeding events (22,23). We added this information in the result section, page 10 line 169.

“In our unit, all patients were treated by anticoagulation therapy with a therapeutic INR range between 2.0 and 2.5. As previously published, we do not use antiplatelet therapy in order to reduce major bleeding events (22,23)”

- In the Discussion (lines 307-310) appears less relevant to the study and can be omitted.

- In the conclusion (lines 344-345)- The authors did not demonstrate that PT directly resulted in renal outcomes and death. Therefore, this statement should be tuned down.

We suppressed this statement accordingly.

References

1. Klootwijk ED, Reichold M, Unwin RJ, Kleta R, Warth R, Bockenhauer D. Renal Fanconi syndrome: taking a proximal look at the nephron. Nephrol Dial Transplant Off Publ Eur Dial Transpl Assoc - Eur Ren Assoc. 2015 Sep;30(9):1456–60. 

2. Peterson PA, Evrin PE, Berggård I. Differentiation of glomerular, tubular, and normal proteinuria: determinations of urinary excretion of beta-2-macroglobulin, albumin, and total protein. J Clin Invest. 1969 Jul;48(7):1189–98. 

3. Lucas GM, Ross MJ, Stock PG, Shlipak MG, Wyatt CM, Gupta SK, et al. Clinical practice guideline for the management of chronic kidney disease in patients infected with HIV: 2014 update by the HIV Medicine Association of the Infectious Diseases Society of America. Clin Infect Dis Off Publ Infect Dis Soc Am. 2014 Nov 1;59(9):e96-138. 

4. Samarawickrama A, Cai M, Smith ER, Nambiar K, Sabin C, Fisher M, et al. Simultaneous measurement of urinary albumin and total protein may facilitate decision-making in HIV-infected patients with proteinuria. HIV Med. 2012 Oct;13(9):526–32. 

5. Walton RJ, Bijvoet OL. Nomogram for derivation of renal threshold phosphate concentration. Lancet Lond Engl. 1975 Aug 16;2(7929):309–10. 

6. Herlitz LC, Mohan S, Stokes MB, Radhakrishnan J, D’Agati VD, Markowitz GS. Tenofovir nephrotoxicity: acute tubular necrosis with distinctive clinical, pathological, and mitochondrial abnormalities. Kidney Int. 2010 Dec;78(11):1171–7. 

7. Gatanaga H, Tachikawa N, Kikuchi Y, Teruya K, Genka I, Honda M, et al. Urinary beta2-microglobulin as a possible sensitive marker for renal injury caused by tenofovir disoproxil fumarate. AIDS Res Hum Retroviruses. 2006 Aug;22(8):744–8. 

8. Loi M. Lowe syndrome. Orphanet J Rare Dis. 2006 May 18;1:16. 

9. Bockenhauer D, Bokenkamp A, van’t Hoff W, Levtchenko E, Kist-van Holthe JE, Tasic V, et al. Renal phenotype in Lowe Syndrome: a selective proximal tubular dysfunction. Clin J Am Soc Nephrol CJASN. 2008 Sep;3(5):1430–6. 

10. Hall AM, Bass P, Unwin RJ. Drug-induced renal Fanconi syndrome. QJM Mon J Assoc Physicians. 2014 Apr;107(4):261–9. 

11. Kormann R, Jacquot A, Alla A, Corbel A, Koszutski M, Voirin P, et al. Coronavirus disease 2019: acute Fanconi syndrome precedes acute kidney injury. Clin Kidney J. 2020 Jun;13(3):362–70. 

12. Salerno CT, Sundareswaran KS, Schleeter TP, Moanie SL, Farrar DJ, Walsh MN. Early elevations in pump power with the HeartMate II left ventricular assist device do not predict late adverse events. J Heart Lung Transplant Off Publ Int Soc Heart Transplant. 2014 Aug;33(8):809–15. 

13. Palevsky PM, Liu KD, Brophy PD, Chawla LS, Parikh CR, Thakar CV, et al. KDOQI US Commentary on the 2012 KDIGO Clinical Practice Guideline for Acute Kidney Injury. Am J Kidney Dis. 2013 May 1;61(5):649–72. 

14. Schaer DJ, Buehler PW, Alayash AI, Belcher JD, Vercellotti GM. Hemolysis and free hemoglobin revisited: exploring hemoglobin and hemin scavengers as a novel class of therapeutic proteins. Blood. 2013 Feb 21;121(8):1276–84. 

15. Silva J-LD, Zand BA, Yang LM, Sabaawy HE, Lianos E, Abraham NG. Heme oxygenase isoform-specific expression and distribution in the rat kidney. Kidney Int. 2001 Apr 1;59(4):1448–57. 

16. Christensen EI, Nielsen R. Role of megalin and cubilin in renal physiology and pathophysiology. Rev Physiol Biochem Pharmacol. 2007;158:1–22. 

17. Chung H-J, Chung J-W, Yi J, Hur M, Lee TH, Hwang S-H, et al. Automation of Harboe method for the measurement of plasma free hemoglobin. J Clin Lab Anal. 2020 Jun;34(6):e23242. 

18. Harboe M. A method for determination of hemoglobin in plasma by near-ultraviolet spectrophotometry. Scand J Clin Lab Invest. 1959;11:66–70. 

19. Muslem R, Caliskan K, Akin S, Sharma K, Gilotra NA, Brugts JJ, et al. Pre-operative proteinuria in left ventricular assist devices and clinical outcome. J Heart Lung Transplant. 2018 Jan 1;37(1):124–30. 

20. Asleh R, Schettle S, Briasoulis A, Killian JM, Stulak JM, Pereira NL, et al. Predictors and Outcomes of Renal Replacement Therapy After Left Ventricular Assist Device Implantation. Mayo Clin Proc. 2019;94(6):1003–14. 

21. Chawla LS, Eggers PW, Star RA, Kimmel PL. Acute Kidney Injury and Chronic Kidney Disease as Interconnected Syndromes. N Engl J Med. 2014 Jul 3;371(1):58–66. 

22. Litzler P-Y, Smail H, Barbay V, Nafeh-Bizet C, Bouchart F, Baste J-M, et al. Is anti-platelet therapy needed in continuous flow left ventricular assist device patients? A single-centre experience. Eur J Cardio-Thorac Surg Off J Eur Assoc Cardio-Thorac Surg. 2014 Jan;45(1):55–9; discussion 59-60. 

23. Netuka I, Litzler P-Y, Berchtold-Herz M, Flecher E, Zimpfer D, Damme L, et al. Outcomes in HeartMate II Patients With No Antiplatelet Therapy: 2-Year Results From the European TRACE Study. Ann Thorac Surg. 2017 Apr;103(4):1262–8.

---

## [Decision Letter · Decision Letter 1]

12 Nov 2020

Hemolysis induced by Left Ventricular Assist Device is associated with proximal tubulopathy.

PONE-D-20-15799R1

Dear Dr. de Nattes,

We’re pleased to inform you that your manuscript has been judged scientifically suitable for publication and will be formally accepted for publication once it meets all outstanding technical requirements.

Kind regards,

Vakhtang Tchantchaleishvili

Academic Editor

PLOS ONE

Additional Editor Comments (optional):

Reviewers' comments:

Reviewer's Responses to Questions

**Comments to the Author**

1. If the authors have adequately addressed your comments raised in a previous round of review and you feel that this manuscript is now acceptable for publication, you may indicate that here to bypass the “Comments to the Author” section, enter your conflict of interest statement in the “Confidential to Editor” section, and submit your "Accept" recommendation.

6. Review Comments to the Author

Reviewer #1: The manuscript has improved substantially. The authors have addressed all the questions raised. I have no further comments.

Reviewer #3: The Authors should be congratulated for their effort to address all the points raised by the reviewers and provide appropriate explanation. The manuscript looks much better and good for publication.

Reviewer #4: All comments have been appropriately addressed. Many concerns were inherent limitations in the dataset. These were mentioned appropriately in the manuscript.

Reviewer #5: The revised manuscript has improved. The authors have addressed my comments appropriately.

I have no further comments.

4. Have the authors made all data underlying the findings in their manuscript fully available?

---

## [Editor Report · Acceptance letter]

16 Nov 2020

PONE-D-20-15799R1 

Hemolysis induced by Left Ventricular Assist Device is associated with proximal tubulopathy. 

Dear Dr. de Nattes:

I'm pleased to inform you that your manuscript has been deemed suitable for publication in PLOS ONE. Congratulations! Your manuscript is now with our production department. 

Kind regards, 

on behalf of

Dr. Vakhtang Tchantchaleishvili 

Academic Editor

PLOS ONE